# Meat Intake, Cooking Methods, Doneness Preferences and Risk of Gastric Adenocarcinoma in the MCC-Spain Study

**DOI:** 10.3390/nu14224852

**Published:** 2022-11-16

**Authors:** Elena Boldo, Nerea Fernández de Larrea, Marina Pollán, Vicente Martín, Mireia Obón-Santacana, Marcela Guevara, Gemma Castaño-Vinyals, Jose María Canga, Beatriz Pérez-Gómez, Inés Gómez-Acebo, Guillermo Fernández-Tardón, Mercedes Vanaclocha-Espi, Rocío Olmedo-Requena, Juan Alguacil, Maria Dolores Chirlaque, Manolis Kogevinas, Nuria Aragonés, Adela Castelló

**Affiliations:** 1Cancer and Environmental Epidemiology Unit, Department of Epidemiology of Chronic Diseases, National Center for Epidemiology, Instituto de Salud Carlos III, Calle de Melchor Fernández Almagro, 5, 28029 Madrid, Spain; 2Consortium for Biomedical Research in Epidemiology & Public Health (CIBER Epidemiología y Salud Pública—CIBERESP), Av. de Monforte de Lemos, 3-5, 28029 Madrid, Spain; 3The Research Group in Gene—Environment and Health Interactions (GIIGAS), Institute of Biomedicine (IBIOMED), Campus Universitario de Vegazana, Universidad de León, 24071 León, Spain; 4Faculty of Health Sciences, Department of Biomedical Sciences, Area of Preventive Medicine and Public Health, Campus Universitario de Vegazana, Universidad de León, 24071 León, Spain; 5Unit of Biomarkers and Susceptibility, Oncology Data Analytics Program, Catalan Institute of Oncology (ICO), Hospital Duran i Reynals, Avinguda de la Gran Via de l’Hospitalet 199-203, 08908 L’Hospitalet de Llobregat, Spain; 6Colorectal Cancer Group, ONCOBELL Program, Bellvitge Biomedical Research Institute (IDIBELL), Avinguda de la Gran Via de l’Hospitalet 199, 08908 L’Hospitalet de Llobregat, Spain; 7Navarra Public Health Institute, Calle Leyre, 15, 31003 Pamplona, Spain; 8Navarra Institute for Health Research (IdiSNA), Calle Leyre 15, 31003 Pamplona, Spain; 9Institute of Global Health (ISGlobal), Carrer del Rosselló 132, 08036 Barcelona, Spain; 10Campus del Mar, Universitat Pompeu Fabra (UPF), Carrer del Dr. Aiguader 80, 08003 Barcelona, Spain; 11IMIM (Hospital del Mar Medical Research Institute), Carrer del Dr. Aiguader 88, 08003 Barcelona, Spain; 12Servicio de Cirugía General y Aparato Digestivo, Complejo Asistencial Universitario de León, 24001 León, Spain; 13Instituto de Investigación Sanitaria Valdecilla, Universidad de Cantabria, Avenida Cardenal Herrera Oria s/n, 39011 Santander, Spain; 14Instituto Universitario de Oncología (IUOPA), Facultad de Medicina, Campus de El Cristo B, Universidad de Oviedo, 33006 Oviedo, Spain; 15Instituto de Investigación Sanitaria del Principado de Asturias (ISPA), Av. Roma s/n, 33011 Oviedo, Spain; 16Cancer and Public Health Area, Foundation for the Promotion of Health and Biomedical Research of Valencia Region (FISABIO), Avda. Catalunya 21, 46020 Valencia, Spain; 17Department of Preventive Medicine and Public Health, School of Medicine, University of Granada, Av. de la Investigación 11, 18016 Granada, Spain; 18Instituto de Investigación Biosanitaria ibs.GRANADA, Doctor Azpitarte 4 4ª Planta, Edificio Licinio de la Fuente, 18012 Granada, Spain; 19Centro de Investigación en Recursos Naturales, Salud y Medio Ambiente, Campus Universitario de El Carmen, Universidad de Huelva, 21071 Huelva, Spain; 20Department of Epidemiology, Regional Health Council, IMIB-Arrixaca, Campus de Ciencias de la Salud, Murcia University, 30120 El Palmar, Spain; 21Public Health Division, Department of Health of Madrid, C/San Martín de Porres, 6, 28035 Madrid, Spain; 22School of Medicine, University of Alcalá, Av. de Madrid, Km 33,600, 28871 Alcalá de Henares, Spain

**Keywords:** stomach neoplasms, red meat, processed meat, cooking methods, doneness preference

## Abstract

Background: The association of meat intake with gastric adenocarcinoma is controversial. We examined the relation between white, red, and processed meat intake and gastric adenocarcinoma, considering doneness preference and cooking methods, by histological subtype and anatomical subsite. Methods: MCC-Spain is a multicase–control study that included 286 incident gastric adenocarcinoma cases and 2993 controls who answered a food-frequency questionnaire. The association of gastric adenocarcinoma with meat intake, doneness preference and cooking methods was assessed using binary multivariate logistic regression mixed models and a possible interaction with sex was considered. Multinomial logistic regression models were used to estimate risk by tumor subsite (cardia vs. non-cardia) and subtype (intestinal vs. diffuse). Sensitivity analyses were conducted comparing models with and without data on *Helicobacter pylori* infection. Results: The intake of red and processed meat increased gastric adenocarcinoma risk (OR for one serving/week increase (95% CI) = 1.11 (1.02;1.20) and 1.04 (1.00;1.08), respectively), specifically among men and for non-cardia and intestinal gastric adenocarcinoma. Those who consume well done white or red meat showed higher risk of non-cardia (white: RRR = 1.57 (1.14;2.16); red: RRR = 1.42 (1.00;2.02)) and intestinal tumors (white: RRR = 1.69 (1.10;2.59); red: RRR = 1.61 (1.02;2.53)) than those with a preference for rare/medium doneness. Stewing and griddling/barbequing red and white meat, and oven baking white meat, seemed to be the cooking methods with the greatest effect over gastric adenocarcinoma. The reported associations remained similar after considering *Helicobacter pylori* seropositivity. Conclusions: Reducing red and processed meat intake could decrease gastric adenocarcinoma risk, especially for intestinal and non-cardia tumors. Meat cooking practices could modify the risk of some gastric cancer subtypes.

## 1. Introduction

Gastric cancer (GC) is the fifth most common cancer worldwide [1,2]. Despite some improvements in the diagnosis and treatment, 5-year survival rate is still lower than 30% in most countries in the world [3,4]. Fortunately, its incidence has been decreasing globally, approaching levels of a rare disease in some populations [5]. This decline has been attributed to changes in the prevalence of exposure to known risk factors for this pathology, associated with higher standards of hygiene, improvements in diet quality and food preservation, increased intake of fresh fruits and vegetables, decreased prevalence of *Helicobacter pylori* (*H. pylori*) infection, and interventions to control tobacco or alcohol consumption, among others [6,7,8]. Globally, geographic differences in the occurrence of this neoplasm have been documented [1]. In Spain, GC mortality also displays a striking geographic pattern [6], and this tumor represents the sixth cause of cancer death [7,8].

Gastric adenocarcinoma (GAC) is the most common histological type of GC (90%). Even though chronic *H. pylori* infection is the main known cause, GC etiology is multifaceted, and diverse risk factors have been associated with this neoplasm, including environmental and genetic factors [9,10]. Some of them, e.g., age, male sex, or GAC family history are not modifiable [11], while others, such as tobacco consumption, diet, or excessive alcohol drinking, are potentially modifiable [12]. Certain contextual and environmental factors are also associated with the development of GC, including parental socioeconomic status, water pollution, soil pollution or soil element content [13].

GAC can be further classified into two histological subtypes according to Lauren’s classification, i.e., intestinal (well-differentiated) and diffuse (undifferentiated) adenocarcinoma [12]. These subtypes differ in their microscopic and gross appearance, sex ratio, age at diagnosis, epidemiologic features, pathogenesis, and prognosis (8). The intestinal type, which accounts for approximately 54% of cases, is more common in elderly males, and shows a slower progression, whereas the diffuse type is more common in younger individuals and has a worse prognosis [14].

Anatomical location of the tumor also has implications in terms of etiology [15]. In recent decades, incidence of cardia gastric adenocarcinoma (CGA) has increased in some geographic areas, while non-cardia gastric adenocarcinoma (NCGA) has decreased [11,16,17]. The main recognized risk factor for NCGA is *H. pylori* chronic infection, and dietary factors are also involved [11], while CGA is more related to gastroesophageal reflux disease and obesity [15,16].

Dietary habits have long been considered as an important risk factor for GAC [10,18,19]. A low intake of fruits, vegetables (except starchy ones), and pulses, as well as a high consumption of salt, salted and smoked foods, chili pepper, processed and grilled/barbequed meats, alcohol, or a high adherence to the Western dietary pattern, have been associated with an increased risk of GAC [20,21]. In terms of meat, its worldwide intake has being steadily rising since the second half of the twentieth century [22]. Total, red, and processed meat intake has been associated with an increased risk of GAC in several studies [23,24,25,26,27,28], and potential underlying biological mechanisms have been identified for this association [23], but evidence is still limited [27,29]. Moreover, meat cooking practices and doneness preferences, which have been less studied, could independently increase the risk of GAC and might help in explaining the heterogeneity currently observed among results of epidemiological studies [23,25,30,31].

In this work, we aimed to elucidate the role of meat intake in the incidence of GAC, including type of meat, meat cooking methods and doneness preferences, overall and by histological subtypes and anatomical subsites, within the MCC-Spain multicase–control study.

## 2. Materials and Methods

### 2.1. Study Population

The MCC-Spain study was carried out between 2008 and 2013 with the objective of identifying etiological factors associated with breast, prostate, colorectal, gastric, and chronic lymphocytic leukemia. The design of the study has already been described [32]. Briefly, in 23 hospitals from 12 Spanish provinces cases of these tumors were prospectively recruited. To ensure a frequency matching by age and sex with the total distribution of cases in each province, controls were chosen at random from the general practitioners’ lists in the same regions. The response rate was 55% among gastric cancer cases and 53% among controls of the full sample of controls (77% among the selected sample of controls for the specific analysis of GC). Two of the provinces did not participate in the recruitment of gastric cancer cases. Participants were invited to join the study if they could complete the questionnaire, were 20 to 85 years old, and had lived in the study area for at least 6 months before the diagnosis/interview. Telephone calls were made to contact controls and those who agreed to participate attended a personal interview. Cases were invited to participate, as promptly as possible after diagnosis was made and were interviewed at their hospital. Histologically confirmed GAC incident cases with no history of the disease were included. Tumors were divided into cardia and non-cardia categories based on their location, and into intestinal and diffuse based on their tumor morphology following the Lauren’s classification [12]. The Ethics Committees of all the participating institutions approved MCC-Spain protocol. All participants signed an informed consent after being informed about the study objectives.

### 2.2. Data Collection and Diet Assessment

In a face-to-face interview, trained personnel administered a structured computerized epidemiological questionnaire to collect data on socio-demographic variables, personal/family medical history and lifestyle, among others. A 154-items semi-quantitative food frequency questionnaire (FFQ) based on a validated instrument [33] was used to assess dietary intake. Height and weight were self-reported. At the end of the interview, the participants received the FFQ and instructions to complete it at home or while in the hospital, and to send the filled form by mail. This FFQ also gathered information on meat cooking methods (griddled-grilled-barbequed, pan-fried/breaded-coated fried, stewed or oven-baked) and used pictures to determine doneness preferences (rare, medium, well-done). The consumption of total, white, red, and processed meat, as well as the consumption of red and white meat according to the frequency of use of each cooking method were reported in servings/week and then converted into grams/day, using sex-specific standard portion sizes. Chicken, turkey, duck, rabbit, and other game were classified as white meat while pork, beef, lamb, liver of any other animal entrails, as well as pork or beef hamburgers or meatballs were considered as red meat. Finally, bacon, sausages, paté/foie-gras, serrano ham and other cold meat were included in the processed meat category. The time reference of the FFQ was the year before cancer diagnosis or before interview in cases and controls, respectively.

Additionally, blood samples were donated by the 61% of cases and 64% of controls who accepted to do it. Samples were processed, aliquoted and stored at −80 °C in the first 48 h. Later, for the *H. pylori* multiplex serology assay, a serum aliquot from each participant was sent on dry ice to be analyzed at the German Cancer Research Centre (DKFZ), in Germany.

### 2.3. Statistical Analyses

Sociodemographic, anthropometric, and lifestyle characteristics were described separately for cases and controls, as well as by type of meat consumption, doneness preference, and cooking methods using basic descriptive statistics. Continuous variables with a normal distribution were described with the mean and the standard deviation (mean -SD-), and differences between cases and controls were tested using t-tests. For continuous variables that presented skewedness, median, and interquartile interval (median (IQI)) were used for description and rank-sum tests to assess differences between cases and controls. Frequencies and percentages were used to describe categorical variables and chi-square tests to assess differences between groups.

Adjusted associations between meat consumption and GAC, by type (white, red, or processed), doneness (rare-medium or well-done) and cooking methods (griddled-grilled-barbequed, pan-fried, stewed, or oven-baked) were evaluated with binary logistic regression mixed models with a random province-specific intercept. All the odds ratios (OR) and 95% confidence intervals (95% CI) estimated by these models were adjusted by education, sex, age, body mass index (BMI), physical activity (metabolic equivalents (METs)) during the previous year, smoking status, family history of gastric cancer and caloric, alcohol, fruits, salty fish, and olive intakes as potential confounders. Additionally, (1) models evaluating the effect of each type of meat (white, red, and processed) were additionally adjusted by the consumption of other types of meat (e.g., for white meat, models were adjusted by red and processed meat intake); (2) models evaluating the effect of doneness preference were also adjusted by total meat-specific intake; and (3) models evaluating the effect of cooking methods, were also adjusted by other meat specific cooking methods (e.g., for griddle-grilled/barbequed (BBQ) white meat, the models were adjusted by fried, stewed, and oven-baked white meat consumption).

A sensitivity analysis was conducted to determine whether the amount of missing values for *H. pylori* seropositivity (36% among participants with complete information on the rest of the variables of interest) influenced the final results. We compared the outcomes of five models, characterized by different sets of adjustment variables (with or without *H. pylori* seropositivity) and by different subsamples (i.e., all the participants, only participants with data on *H. pylori* serostatus or only *H. pylori* seropositive participants). Since the direction and magnitude of the associations found were similar in all the models, to optimize the statistical power we present the results of the analyses in the main text without considering *H. pylori* related information. Results considering data on *H. pylori* serostatus are shown in the Online Resource 1, Appendix A.

An interaction term between sex and meat consumption (by type, doneness, and cooking method) was included in the models to check for heterogeneity of the effects by sex. Models with and without the interaction term were compared using the likelihood ratio test to obtain the *P* value for heterogeneity.

The same associations were evaluated taking into account tumor location (no cancer; cardia GC; and non-cardia GC) and morphology (no cancer; intestinal GC and diffuse GC) using multinomial logistic regression models that provided relative risk ratio (RRR) and 95% CI.

Meat consumption was analyzed both as a continuous variable and grouping weekly intake in grams into quartiles of their distribution among controls. However, to improve interpretability, results are presented in servings/week, with one serving representing 125 g of total, red, or white meat, and as 50 g of processed meat. For the analyses by sex, weekly intake was grouped using sex-specific data distribution among controls. The analyses for doneness preference and cooking methods were restricted to group specific meat consumers. All analyses were performed with Stata/MP 16.0 (StataCorp, College Station, TX, USA).

## 3. Results

Initially, 459 cases of GC and 3440 controls were enrolled, with 344 (75%) cases and 2993 (87%) controls providing valid dietary information. Cases whose dietary information was collected more than 6 months after diagnosis (*n* = 38), with an uncertain diagnosis date (*n* = 2), or with tumors that were not adenocarcinomas (*n* = 18) were excluded from the analyses. Therefore, the present study included 286 GAC cases and 2993 controls aged 26 to 85 years (Figure 1).

Bivariate analyses showed that cases were primarily men, older than the controls and with lower educational level. GAC cases also showed higher caloric and alcohol intake, lower physical activity, and higher prevalence of family history of gastric cancer (Table 1). In terms of meat consumption, cases showed higher consumption of all types of meat and reported a higher preference for well-done meat, and for pan-fried (red meat), stewed (white and red meat), and oven-baked (white meat) cooking methods (Table 2).

Table 3 shows the association between GAC and total, white, red, and processed meat consumption for the whole population and by sex. Compared to individuals with a total meat intake under 3.0 servings/week, participants with an intake of 4.3–5.9 or over 5.9 servings/week showed a 58% (OR (95% CI) = 1.58 (1.02;2.42)) and a 73% (1.73 (1.10;2.71)) higher risk of GAC, respectively (*p*-trend = 0.01). This was mainly due to the effect of red and, to a lesser extent, processed meat intake. Individuals with a red meat consumption of 1.8–2.9 servings/week or over 2.9 servings/week showed a 73% (OR = 1.73 (1.12;2.66)) and 76% (OR = 1.76 (1.14;2.72)) higher risk of GAC, respectively, than those eating less than 1.1 servings/week. For processed meat, the risk of GAC increased by 4% (OR = 1.04 (1.00;1.08)) for each increase in one serving/week. The corresponding linear increase for total and red meat was 11%. When evaluating the effect separately by sex, the observed associations were mainly among men.

Similarly, analyses by tumor subsite (Table 4) revealed that associations, when present, were observed with total, red, and processed meat. Although no statistically significant heterogeneity was detected, no strong associations were found for cardia adenocarcinoma, whereas non-cardia adenocarcinoma was associated with a consumption of over 4.3 servings/week of total meat (*p*-trend = 0.013). Specifically, individuals with an intake of 4.3–5.9 servings/week showed a 70% (RRR (95% CI) = 1.70 (1.04;2.80)) and those over 5.9 servings/week a 95% (RRR = 1.95 (1.16;3.27)) higher risk of non-cardia adenocarcinoma than those eating meat less than 3.0 times per week. As for red meat, individuals with intakes between 1.8–2.9 and over 2.9 servings/week, showed a 72% (RRR = 1.72 (1.06;2.78)) and a 69% (RRR = 1.69 (1.04;2.76)) higher risk of non-cardia adenocarcinoma, respectively (*p*-trend = 0.023), than those with the lowest consumption (<1.1 servings/week). Finally, non-cardia cancer risk was a 79% higher (RRR = 1.79 (1.10;2.92)) among those eating 4.3 servings/week or more than among those consuming less than 1.4 servings/week of processed meat.

Analyses by tumor morphology (Table 4) showed more striking differences, with markedly stronger associations for intestinal tumors with red meat. Compared with individuals eating less than 1.1 servings/week of red meat, the risk of intestinal GAC was 2.77 (95% CI = 1.28;6.00) times bigger for those consuming between 1.1 and 1.8 servings/week, 3.62 (1.66;7.90) times bigger for 1.8–2.9 servings/week, and 5.60 (2.58;12.13) times bigger for intakes ≥ 2.9 servings/week.

Regarding meat doneness preference and cooking methods (Table 5), our results show an increased risk of GAC for the intake vs. non intake of white meat for all cooking methods, especially for stewing (OR (95% CI) = 1.71 (1.19;2.47)), oven-baking (OR = 1.62 (1.20;2.20)), and griddling/barbequing (OR = 1.49 (1.08;2.07)). By sex, differences were mainly observed among men (stewing: OR = 1.97 (1.26;3.08); oven-baking: OR = 2.03 (1.41;2.93); griddling/barbequing: OR = 1.56 (1.07;2.27)). The intake of stewed (OR (95% CI) = 1.62 (1.01;2.60)) and griddled/barbequed (OR = 1.59 (1.03;2.45)) red meat was also linked to a higher global GAC risk. When considering tumor subsite and morphology (Table 6) and referencing individuals with rare/medium doneness preference, those eating well-done white and red meat presented a 57% (RRR (95% CI) = 1.57 (1.14;2.16)) and a 42% (RRR = 1.42 (1.00;2.02)) higher risk of NCGA, and a 69% (RRR = 1.69 (1.10;2.59)) and 61% (RRR = 1.61 (1.02;2.53)) higher risk of intestinal GAC, respectively. As for the cooking methods, stewing and oven-baking white meat were associated with NCGA (stewed: RRR = 1.73 (1.14;2.63); oven-baked: RRR = 1.36 (0.98;1.88)), with cardia GAC (oven-baked: RRR = 2.14 (1.15–3.96)), and with intestinal tumors (stewed: RRR = 2.40 (1.28;4.49); oven-baked: RRR = 1.57 (1.01;2.44)). Griddling/barbequing red meat was also associated with these tumor subtypes (NCGA: RRR = 1.71 (1.02;2.85); intestinal: RRR = 2.16 (1.09;4.29)).

## 4. Discussion

Our results indicate that meat consumption may increase GAC risk, specifically among men and non-cardia and intestinal tumors, mainly due to the association with red and, to a lesser extent, processed meat. Increased GAC risk was observed for consumption of 1.8 or more 125 g-servings per week of red meat and 4.3 or more 50 g-servings per week of processed meat. Doneness preference might also influence GAC risk. In fact, participants who preferred well-done white or red meat presented a higher risk of non-cardia and intestinal tumors than those with a preference for rare/medium doneness. Additionally, stewing and griddling/barbequing red and white meat and oven baking white meat appeared to be the cooking methods with the greatest effect over GAC.

A positive dose–response association was observed between GAC and both total and red meat and, to a lesser extent, processed meat intake. In 2017, a systematic review found null results in cohort studies for the association between GAC risk with red and processed meat consumption, while case–control studies yielded positive associations [25]. In 2019, a meta-analysis including 43 studies (11 cohort studies and 32 case–control studies), concluded that red and processed meat consumption increased the risk of GAC by 41% and 57%, respectively [23]. More recently, the Stomach Cancer Pooling (StoP) Project consortium reported an elevated risk of gastric cancer for the consumption of total (OR: 1.30; 95% CI: 1.09–1.55), red (OR: 1.24; 95% CI: 1.00–1.53), and processed meat (OR: 1.23; 95% CI: 1.06–1.43) [26]. With regard to white meat intake, a reverse association with the risk of this malignancy has been found in some studies [23,34], but not in all [20,24]. In our study, consumption of white meat had no clear effect on GAC, although it was associated with higher risk of diffuse GAC, and specific cooking methods were also associated with increased risk. Variability in these aspects could underlie the inconsistent results found among published studies.

Regarding results by anatomical subsite, in agreement with our findings, in the European Prospective Investigation Into Cancer and Nutrition (EPIC) cohort, total, red, and processed meat consumption was associated with an elevated risk of non-cardia gastric adenocarcinoma, especially in *H. pylori* antibody-positive individuals, while not with cardia adenocarcinoma [24]. By histological subtype, apart from the EPIC study, we have only found a previous study, conducted in Italy, which studied the risk patterns for intestinal and diffuse adenocarcinoma subtypes [35]. In these studies, high meat intake was associated with an increased risk for both subtypes, with no remarkable differences between them. In our study, total and red meat intake were strongly and positively associated with the risk of having an intestinal type tumor, with a clear dose–response effect, while no association was observed for diffuse GAC.

When comparing the highest and lowest categories of total meat intake separately by sex, a positive association was observed mainly among men, mostly due to the effect of red and processed meat. Additionally, in the dose–response analysis, a linear increase for total, red, and processed meat was observed among men. These results support that consumption of processed meat increases the risk of gastric adenocarcinoma. However, the number of GAC cases among women was small, due to the lower incidence of this tumor among women, what resulted in less precise estimates. More research is warranted to explore possible differences in the effects of meat intake in gastric cancer between women and men.

Several studies have explored the biological mechanism that could explain the association between red and processed meat ingestion and GC. First, it has been hypothesized that high dietary iron intake and elevated body iron status increase the risk of several cancers, including GAC [36,37,38]. Iron contained in red and processed meat may cause oxidative DNA damage by catalyzing the formation of reactive oxygen species, and involving the endogenous formation of carcinogenic N-nitroso compounds (NOCs) [39]. In addition, heme iron is considered a critical factor for bacterial growth, such as *H. pylori*, the main established risk factor of non-cardia gastric cancer [40,41]. Furthermore, independently of iron, consumption of processed foods contributes to an increased intake of salt, saturated fats, cholesterol, polycyclic aromatic hydrocarbons (PAHs), and heterocyclic amines (HCAs), compounds that have also been described as potential carcinogens [42], that may affect different intracellular pathways related to proliferation, angiogenesis, inflammatory responses, or apoptosis [43]. Salt irritates the gastric mucosa rendering it more susceptible to chemical carcinogenesis and to *H. pylori* colonization [19,44]. As for white meat, compared to red meat, it contains less cholesterol, less saturated fats, and also a lower content of heme iron, reducing the endogenous formation of NOCs [45]. Moreover, white meat is a source of polyunsaturated fatty acids (PUFAs), some of which possess anti-inflammatory activity and induce apoptosis [46]. Thus, white meat intake could contribute to decreasing the risk of gastric adenocarcinoma by limiting chronic mucosal inflammation, which is a major risk factor for gastric carcinogenesis. Nevertheless, residual confounding cannot be completely discarded, since overall healthier dietary patterns might be associated with higher intake of white meat [47,48,49].

The methods used to cook, process, and preserve meat may also influence the risk of GAC. According to our findings, an increased risk of NCGA and intestinal tumors was observed among consumers of well-done white or red meat. Moreover, stewing and oven-baking seemed to be the cooking methods with the greatest effects over GAC for white meat. Griddling/barbequing and stewing red meat were also associated with GAC. A previous case–control study [30] exploring the association between different methods of cooking meat (roasting/grilling; boiling/stewing; frying/pan frying) and cancer risk, also reported a statistically significant association with gastric cancer risk only for boiled/stewed red meat (OR = 1.86; 95% CI: 1.20–2.87). Depending on the type of meat, cooking method, and level of doneness, red and white meats cooked at high temperature or for a long period of time (e.g., griddling, barbequing, stewing) may form high levels of mutagens, including HCAs and PAHs, which could have an important role in GAC pathogenesis [50,51,52,53,54,55]. Stewed dishes might have higher amounts of these carcinogens, since these substances may remain in the sauce [56]. The use of cooking method or doneness preference as a measure of exposure, rather than individual compounds, may reflect the carcinogenicity of all known and unknown elements present in cooked meat.

Although many factors contribute to the development of gastric cancer, the dominant risk factor for NCGC is chronic infection with *H. pylori* [57,58]. However, adjusting our models by *H. pylori* antibody status had no significant effect in the results (Online Resource 1, Appendix A). *H. pylori* infection is highly prevalent worldwide, and, in our study, 89% of participants were seropositive [59]. With this high prevalence of infection, there was insufficient heterogeneity to observe a possible confounding or modification effect of infection status on the association between GAC and meat consumption.

Several limitations should be considered before drawing conclusions. Firstly, as is always the case with case–control studies, particularly when assessing the effect of self-reported dietary data, we are concerned about the possible recall bias. Anticipating the possible presence of this bias, we included in the questionnaire specific questions on general eating habits that were subsequently used to adjust the responses of the FFQ [60], and we excluded all cases that answered the questionnaire more than 6 months after diagnosis. Moreover, carcinogenesis may be associated with long-term lifestyles, and the ability of our FFQ to assess long-term dietary habits may be of concern. However, several studies have found a strong agreement between recall of past diet and current diet, suggesting that if remote diet is of interest, focusing questions on the period of interest generally provides appropriate information up to approximately 10 years [61]. Recall bias can also affect self-reported height and weight information, which would influence the calculation of the BMI in the year prior to GC diagnosis or interview. Additionally, participation rate was 55% for cases and 53% for controls, and though it could appear to be low, there is general consensus that a 50% rate might be adequate [62], especially when biological samples are collected.

As for the representativeness of the meat intake collected between 2008 and 2013, meat consumption has decreased in Spain in the last 10 years, especially for red and processed meat [63]. However, this does not reduce the validity of the results obtained in this study. Cut points are relative, therefore, general changes in meat consumption would result in different cut points for quartiles but similar distribution of individuals and therefore similar associations. Differences in the estimates would be observed if changes occur differently among cases and controls. In this case, we would expect the largest decrease in meat intake to occur among controls, which would lead to stronger effects. Moreover, the main results were not adjusted by *H. pylori* infection. However, the sensitivity analyses performed (Appendix A) did not show differences in the effect estimates when considering this factor, thus supporting that the main results are not confounded by *H. pylori* infection status. Finally, since cases and controls are frequency matched for the whole sample that includes all five cancer locations investigated in the MCC-Spain study, some imbalance in the characteristics of cases and controls are observed when analyzing one specific tumor. However, to correct the possible bias introduced by this misbalance, all multivariable analyses were adjusted by age and sex, as well as for gastric cancer risk factors.

Our study also has some notable strengths. Incident GAC cases were recruited, and all of them had histologic confirmation. We were able to analyze different subgroups, according to tumor location and histological subtype, which have been scarcely addressed in previous studies on meat intake and gastric cancer, and to test interactions with sex. We collected information on main possible confounders, such as BMI, alcohol and tobacco consumption, physical activity, and family history of gastric cancer, and all our estimates were adjusted for these factors. Another important asset is the use of a detailed FFQ to collect information on type of meat consumption, doneness preference, and cooking methods, which also included pictures to ensure the correct classification of doneness preference. In addition, the recruitment of cases and controls in 10 provinces from the North, South, Centre, West, and East of the country allowed us to capture the geographical variability in the consumption of meat in Spain.

## 5. Conclusions

This work supports the hypothesis of an association between the intake of different types of meat and GAC. A differential role of doneness preferences and cooking methods in this relationship was also suggested. Reducing red and processed meat consumption and avoiding overcooking and preparing meat at high temperatures for long periods of time, could help to reduce the risk of GAC, especially intestinal and non-cardia tumors. Further research is warranted to reinforce the strength of the evidence and to improve our understanding of the possible underlying mechanisms.

## Figures and Tables

**Figure 1 nutrients-14-04852-f001:**
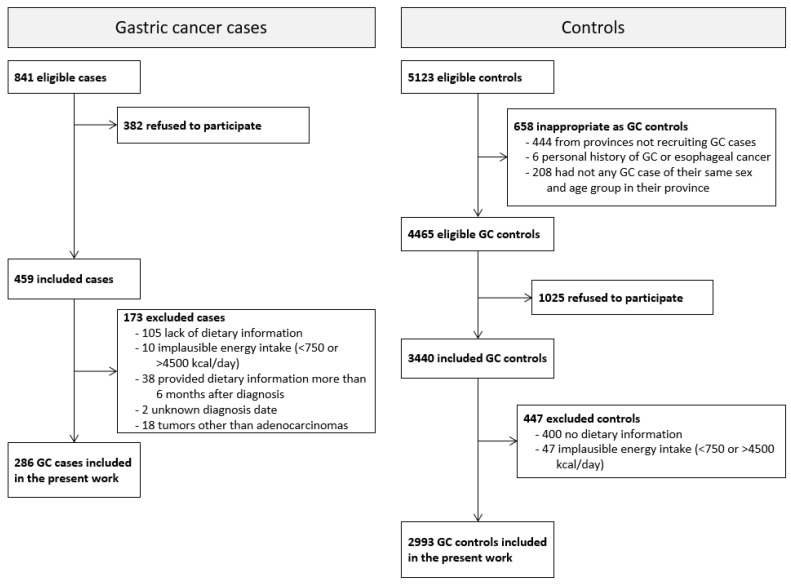
Selection process of gastric cancer cases and controls in the MCC-Spain study from 2008 to 2013.

**Table 1 nutrients-14-04852-t001:** Description of socio-demographic and other baseline characteristics of controls and gastric adenocarcinoma cases in the MCC-Spain study.

	Controls	Cases	*p*-Value
	*n* = 2993	*n* = 286	
**Age (years), mean (SD)**	63.94 (11.42)	66.31 (12.40)	0.001
**Age (10-year periods), *n* (%)**			<0.001
26–35	28 (1%)	6 (2%)	
36–45	195 (7%)	10 (4%)	
46–55	462 (15%)	44 (15%)	
56–65	837 (28%)	64 (22%)	
66–75	975 (33%)	84 (29%)	
76–85	496 (17%)	778 (27%)	
**Education, *n* (%)**			<0.001
No formal Education	537 (18%)	80 (28%)	
Primary School	1007 (34%)	110 (38%)	
Secondary School	837 (28%)	65 (23%)	
University or higher	612 (20%)	31 (11%)	
**Sex, *n* (%)**			<0.001
Male	1669 (56%)	200 (70%)	
Female	1324 (44%)	86 (30%)	
**Energy (kcal/day),** **mean (SD)**	1912.50 (571.94)	2095.19 (651.32)	<0.001
**Alcohol(g/day),** **median (IQI)**	7.57 (0.00;24.72)	12.92 (1.41;40.42)	<0.001
**Salty fish (g/day),** **median (IQI)**	0.00 (0.00;4.59)	0.00 (0.00;4.59)	0.973
**Olives (g/day),** **median (IQI)**	1.97 (0.00;6.42)	1.97 (0.00;6.42)	0.534
**Fruits(g/day),** **median (IQI)**	330.21 (195.32;460.65)	332.53 (203.02;456.81)	0.947
**BMI ^a^ (kg/m^2^),** **median (IQI)**	26.68 (4.33)	27.15 (3.86)	0.084
**Physical activity (METs/week) ^a^, *n* (%)**			0.001
0	1169 (40%)	141 (49%)	
0.1–8	406 (14%)	33 (12%)	
8.1–16	351 (12%)	15 (5%)	
>16	1029 (35%)	97 (34%)	
**Smoking ^a^, *n* (%)**			0.708
Never Smoker	1313 (44%)	120 (42%)	
Former Smoker	1076 (36%)	103 (36%)	
Current Smoker	592 (20%)	62 (22%)	
**Family history of GC, *n* (%)**			<0.001
No	2661 (89%)	223 (78%)	
2nd degree	139 (5%)	14 (5%)	
One of 1st degree	181 (6%)	42 (15%)	
More than one of 1st degree	12 (0%)	7 (2%)	
**Serology against *H. pylori* ^a^, *n* (%)**			0.117
Negative	218 (11%)	13 (7%)	
Positive	1701 (89%)	161 (93%)	

GC: gastric cancer; IQI: interquartile interval; SD: standard deviation. ^a^ BMI (body mass index) was missing for 139 (4.2%) participants, 125 (4.9%) controls and 14 cases (4.2%); physical activity was missing for 38 (1.2%) participants, all of them controls; smoking was missing for 13 (0.4%) participants, 12 (0.4%) controls and 1 case (0.4%); and *H. pylori* test was not available for 1186 (36.2%) individuals, 1074 (35.9%) controls and 112 cases (39.2%).

**Table 2 nutrients-14-04852-t002:** Description of meat intake, doneness preference and cooking methods for controls and gastric adenocarcinoma cases in the MCC-Spain study.

	Controls	Cases	*p*-Value
	*n* = 2993	*n* = 286	
**Daily Intake**			
** Total Meat**			
Non-Consumers, *n* (%)	7 (0.23%)	0 (0.00%)	
Intake (g/day), median ^a^ (IQI)	76.11 (53.92;105.57)	89.83 (66.38;129.64)	<0.001
** White Meat**			
Non-Consumers, *n* (%)	115 (3.84%)	2 (0.70%)	
Intake (g/day), median ^a^ (IQI)	19.22 (12.94;27.23)	21.32 (15.37;32.05)	<0.001
** Red Meat**			
Non-Consumers, *n* (%)	83 (2.77%)	2 (0.70%)	
Intake (g/day), median ^a^ (IQI)	32.21 (19.05;51.01)	40.97 (25.85;61.20)	<0.001
** Processed Meat**			
Non-Consumers, *n* (%)	82 (2.74%)	5 (1.75%)	
Intake (g/day), median ^a^ (IQI)	18.51 (9.91;30.35)	23.29 (13.29;40.68)	<0.001
**Doneness Preference, *n* (%)**			
** White Meat ^b^**			0.006
Rare	196 (7.32%)	10 (3.89%)	
Medium	1629 (60.87%)	143 (55.64%)	
Well-done	851 (31.80%)	104 (40.47%)	
** Red Meat ^b^**			0.012
Rare	286 (10.42%)	28 (10.53%)	
Medium	1894 (69.02%)	163 (61.28%)	
Well-done	564 (20.55%)	75 (28.20%)	
**Cooking Method, median ^a^ (IQI)**			
** White Meat (g/day)**			
Griddle-grilled/BBQ	3.99 (0.00;8.96)	3.77 (0.00;7.83)	0.246
Pan-Fried/breaded-coated fried	1.90 (0.00;5.73)	2.61 (0.00;6.51)	0.088
Stewed	3.42 (0.00;7.22)	4.97 (1.85;9.37)	<0.001
Oven-baked	1.58 (0.00;3.88)	2.40 (0.00;4.74)	0.001
Other/Unknown	0.00 (0.00;0.00)	0.00 (0.00;0.00)	0.287
** Red Meat (g/day)**			
Griddle-grilled/BBQ	11.44 (3.63;21.75)	11.71 (3.71;22.44)	0.726
Pan-Fried/breaded-coated fried	3.31 (0.00;11.20)	5.68 (0.00;14.03)	0.002
Stewed	8.27 (3.03;15.17)	10.91 (4.85;21.02)	<0.001
Oven-baked	0.00 (0.00;3.19)	0.00 (0.00;3.32)	0.617
Other/Unknown	0.00 (0.00;1.64)	0.00 (0.00;2.05)	<0.001

^a^ Median includes non-consumers. ^b^ Doneness preference for white meat was missing for 220 (6.98%) participants, 193 (6.73%) controls and 27 (9.51%) cases and doneness preference for red meat was missing for 132 (4.20%) participants, 116 (4.06%) controls and 16 (5.67%) cases. BBQ: barbequed; IQI: interquartile interval.

**Table 3 nutrients-14-04852-t003:** Association between gastric adenocarcinoma incidence and total, white, red, and processed meat consumption overall and by sex.

		All						Males						Females					
	Sv/Week ^a^	Controls	Cases	OR ^b^	LL ^b^	UL ^b^	Sv/Week ^a^	Controls	Cases	OR ^b^	LL ^b^	UL ^b^	Sv/Week ^a^	Controls	Cases	OR ^b^	LL ^b^	UL ^b^	*p*–Het ^b^
**Weekly Intake**		***n* = 2821**	***n* = 271**					***n* = 1587**	***n* = 196**					***n* = 1234**	***n* = 75**				
** Total Meat ^c^**																			
Q1 ^d^	<3.0	707	38	1			<3.44	396	27	1			<2.61	314	12	1			**0.004**
Q2 ^d^	3.0–4.3	700	55	1.28	0.82	1.99	3.44–4.81	400	39	1.16	0.68	1.98	2.61–3.61	305	16	1.37	0.63	2.99	
Q3 ^d^	4.3–5.9	716	75	**1.58**	**1.02**	**2.42**	4.81–6.55	403	43	1.37	0.81	2.32	3.61–5.03	309	33	**2.67**	**1.33**	**5.36**	
Q4 ^d^	≥5.9	698	103	**1.73**	**1.10**	**2.71**	≥6.55	388	87	**2.39**	**1.44**	**3.98**	≥5.03	306	14	1.06	0.47	2.40	
*p*–trend				**0.012**						**<0.001**						0.409			
1 serving/week increase				**1.11**	**1.05**	**1.17**				**1.13**	**1.06**	**1.20**				1.02	0.91	1.14	0.095
** White Meat ^e^**																			
Q1 ^d^	<0.7	709	49	1			<0.72	400	34	1			<0.72	308	15	1			0.926
Q2 ^d^	0.7–1.1	716	62	1.24	0.82	1.88	0.72–1.19	399	54	1.28	0.79	2.07	0.72–0.95	321	21	1.25	0.62	2.54	
Q3 ^d^	1.1–1.5	698	72	1.05	0.70	1.57	1.19–1.56	391	43	0.97	0.59	1.60	0.95–1.39	306	17	0.95	0.45	1.99	
Q4 ^d^	≥1.5	698	88	1.38	0.93	2.04	≥1.56	397	65	1.43	0.89	2.29	≥1.39	299	22	1.31	0.65	2.64	
*p*–trend				0.195						0.269						0.641			
1 serving/week increase				1.11	0.98	1.26				1.14	0.99	1.31				1.01	0.76	1.35	0.475
** Red Meat ^e^**																			
Q1 ^d^	<1.1	710	39	1			<1.25	404	31	1			<0.91	309	14	1			0.371
Q2 ^d^	1.1–1.8	700	58	1.39	0.89	2.15	1.25–2.07	387	35	1.04	0.61	1.76	0.91–1.50	308	17	1.09	0.52	2.31	
Q3 ^d^	1.8–2.9	699	76	**1.73**	**1.12**	**2.66**	2.07–3.18	399	58	**1.65**	**1.01**	**2.70**	1.50–2.44	311	25	1.78	0.89	3.57	
Q4 ^d^	≥2.9	712	98	**1.76**	**1.14**	**2.72**	≥3.18	397	72	**1.72**	**1.06**	**2.81**	≥2.44	306	19	1.24	0.59	2.60	
*p*–trend				**0.009**						**0.008**						0.339			
1 serving/week increase				**1.11**	**1.02**	**1.20**				**1.12**	**1.03**	**1.22**				1.02	0.84	1.23	0.337
** Processed Meat ^e^**																			
Q1 ^d^	<1.4	700	43	1			<1.72	398	34	1			<1.15	307	12	1			0.300
Q2 ^d^	1.4–2.6	706	59	1.25	0.82	1.92	1.72–3.13	398	41	1.04	0.63	1.72	1.15–2.09	316	22	1.63	0.78	3.40	
Q3 ^d^	2.6–4.3	718	67	1.22	0.80	1.87	3.13–5.03	401	47	1.18	0.72	1.93	2.09–3.28	305	19	1.40	0.66	3.00	
Q4 ^d^	≥4.3	697	102	1.48	0.97	2.28	≥5.03	390	74	**1.67**	**1.02**	**2.72**	≥3.28	306	22	1.27	0.60	2.69	
*p*–trend				0.095						**0.026**						0.771			
1 serving/week increase				**1.04**	**1.00**	**1.08**				**1.06**	**1.01**	**1.11**				0.99	0.91	1.08	0.148

^a^ One serving (sv) of total, white, and red meat = 125 g. One serving of processed meat = 50 g. ^b^ OR: odds ratio; LL: lower limit of the 95% confidence interval; UL: upper limit of the 95% confidence interval; *p*-het: *p* value for the heterogeneity of effects between sexes. ^c^ For total meat, binary logistic regression models adjusted by sex (only for “All”), age, education, family history of stomach cancer, physical activity (METs), smoking, BMI and energy, alcohol, fruits, salty fish, and olive intakes as fixed effects and province of residence as a random effect. ^d^ Quartiles calculated among controls. ^e^ For white, red, and processed meat, binary logistic regression models are adjusted by sex (only for “All”), age, education, family history of stomach cancer, physical activity (METs), smoking, BMI, energy, alcohol, fruits, salty fish, and olive intakes, and the consumption of other types of meat as fixed effects and province of residence as a random effect.

**Table 4 nutrients-14-04852-t004:** Association between gastric adenocarcinoma incidence and total, white, red, and processed meat consumption by tumor subsite (cardia, non-cardia) and morphology (intestinal, diffuse).

			Tumor Subsite									Tumor Morphology								
		Controls *n* = 2821	Cardia *n* = 65				Non-Cardia *n* = 199					Intestinal *n* = 106				Diffuse *n* = 66				
	Sv/Week ^a^	*n*	*n*	RRR ^b^	LL ^b^	UL ^b^	*n*	RRR ^b^	LL ^b^	UL ^b^	*p*-Het ^b^	*n*	RRR ^b^	LL ^b^	UL ^b^	*n*	RRR ^b^	LL ^b^	UL ^b^	*p*-Het ^b^
**Weekly Intake**																				
** Total Meat ^c^**																				
Q1 **^d^**	<3.0	707	9	1			28	1			0.444	12	1			11	1			0.119
Q2 **^d^**	3.0–4.3	700	8	0.66	0.24	1.77	45	1.49	0.90	2.46		22	1.74	0.83	3.66	17	1.36	0.62	2.99	
Q3 **^d^**	4.3–5.9	716	20	1.20	0.52	2.78	54	**1.70**	**1.04**	**2.80**		33	**2.80**	**1.37**	**5.70**	13	0.92	0.39	2.13	
Q4 **^d^**	≥5.9	698	28	1.04	0.43	2.48	72	**1.95**	**1.16**	**3.27**		39	**3.23**	**1.53**	**6.84**	25	1.29	0.56	2.95	
*p*-trend				0.576				**0.013**					**0.001**				0.773			
1 serving/week increase				1.08	0.98	1.19		**1.11**	**1.04**	**1.18**	0.651		**1.17**	**1.08**	**1.28**		0.99	0.89	1.10	**0.011**
** White Meat ^e^**																				
Q1 **^d^**	<0.7	709	10	1			38	1			0.578	22	1			7	1			0.057
Q2 **^d^**	0.7–1.1	716	12	1.41	0.58	3.41	49	1.19	0.75	1.88		28	1.08	0.58	2.02	23	**3.06**	**1.26**	**7.41**	
Q3 **^d^**	1.1–1.5	698	23	1.38	0.63	3.04	46	0.89	0.56	1.42		23	0.55	0.29	1.05	19	**2.55**	**1.02**	**6.37**	
Q4 **^d^**	≥1.5	698	20	1.32	0.59	2.98	66	1.39	0.89	2.16		33	0.89	0.48	1.65	17	2.34	0.93	5.91	
*p*-trend				0.592				0.262					0.424				0.212			
1 serving/week increase				1.03	0.81	1.31		1.14	0.99	1.32	0.438		1.05	0.84	1.32		1.13	0.93	1.38	0.626
** Red Meat ^e^**																				
Q1 **^d^**	<1.1	710	6	1			32	1.00			0.908	10	1			16	1			**<0.001**
Q2 **^d^**	1.1–1.8	700	14	1.89	0.70	5.08	43	1.29	0.79	2.11		26	**2.77**	**1.28**	**6.00**	12	0.60	0.27	1.32	
Q3 **^d^**	1.8–2.9	699	18	2.02	0.76	5.35	57	**1.72**	**1.06**	**2.78**		28	**3.62**	**1.66**	**7.90**	22	0.91	0.45	1.85	
Q4 **^d^**	≥2.9	712	27	2.05	0.78	5.37	67	**1.69**	**1.04**	**2.76**		42	**5.60**	**2.58**	**12.13**	16	0.54	0.25	1.19	
*p*-trend				0.223				**0.023**					**<0.001**				0.310			
1 serving/week increase				1.11	0.97	1.26		**1.10**	**1.00**	**1.20**	0.938		**1.26**	**1.12**	**1.42**		0.89	0.74	1.07	**0.002**
** Processed Meat ^e^**																				
Q1 **^d^**	<1.4	700	8	1			32	1.00			0.767	21	1			10	1			0.805
Q2 **^d^**	1.4–2.6	706	14	1.50	0.61	3.72	45	1.34	0.82	2.17		22	0.89	0.47	1.72	13	1.14	0.48	2.67	
Q3 **^d^**	2.6–4.3	718	16	1.23	0.50	3.02	49	1.35	0.83	2.19		26	0.96	0.51	1.81	18	1.58	0.70	3.55	
Q4 **^d^**	≥4.3	697	27	1.29	0.53	3.16	73	**1.79**	**1.10**	**2.92**		37	1.21	0.63	2.31	25	1.78	0.77	4.09	
*p*-trend				0.751				**0.029**					0.539				0.078			
1 serving/week increase				1.03	0.95	1.11		**1.05**	**1.00**	**1.10**	0.605		1.05	0.99	1.12		1.02	0.96	1.10	0.589

^a ^One serving (sv) of total, white, and red meat was a portion of 125 g. One serving of processed meat was a portion of 50 g. ^b^ RRR: relative risk ratio; LL: Lower limit of the 95% confidence interval; UL: Upper limit of the 95% confidence interval; *p*-het: *p* value for the heterogeneity of effects among subtypes. ^c^ For total meat, multinomial logistic regression models are adjusted by sex, age, education, family history of stomach cancer, physical activity (METs), smoking, BMI and energy, alcohol, fruits, salty fish and olives intake and province of residence. ^d^ Quartiles calculated among controls. ^e^ For white, red and processed meat, multinomial logistic regression models are adjusted by sex, age, education, family history of stomach cancer, physical activity (METs), smoking, BMI, energy, alcohol, fruits, salty fish and olives intake, the consumption of other types of meat and province of residence.

**Table 5 nutrients-14-04852-t005:** Association between gastric adenocarcinoma incidence and meat-type specific doneness preference and cooking methods, overall and by sex (restricted to consumers of each type of meat).

	All					Males					Females					
	**Controls**	**Cases**	**OR ^a,b^**	**LL ^a,b^**	**UL ^a,b^**	**Controls**	**Cases**	**OR ^a,c^**	**LL ^a,c^**	**UL ^a,c^**	**Controls**	**Cases**	**OR ^a,c^**	**LL ^a,c^**	**UL ^a,c^**	** *p* ** **-Int ^a^**
**Doneness Preference**																
** White Meat**	** *n* ** **= 2520**	** *n* ** **= 244**				** *n* ** **= 1394**	** *n* ** **= 173**				** *n* ** **= 1126**	** *n* ** **= 71**				
Rare/Medium	1740	145	1			960	98	1			780	47	1			
Well-done	780	99	1.16	0.86	1.56	434	75	1.29	0.90	1.83	346	24	0.91	0.53	1.54	0.268
** Red Meat**	** *n* ** **= 2582**	** *n* ** **= 253**				** *n* ** **= 1464**	** *n* ** **= 183**				** *n* ** **= 1118**	** *n* ** **= 70**				
Rare/Medium	2067	185	1			1169	132	1			898	53	1			
Well-done	515	68	1.23	0.89	1.69	295	51	1.23	0.84	1.80	220	17	1.22	0.68	2.20	0.984
	**Controls**	**Cases**	**OR ^a,d^**	**LL ^a,d^**	**UL ^a,d^**	**Controls**	**Cases**	**OR ^a,e^**	**LL ^a,e^**	**UL ^a,e^**	**Controls**	**Cases**	**OR ^a,e^**	**LL ^a,e^**	**UL ^a,e^**	** *p* ** **-Int ^a^**
**Cooking Methods**																
** White Meat**	** *n* ** **= 2711**	** *n* ** **= 269**				** *n* ** **= 1516**	** *n* ** **= 194**				** *n* ** **= 1195**	** *n* ** **= 75**				
Griddle/BBQ	1961	185	**1.49**	**1.08**	**2.07**	1051	131	**1.56**	**1.07**	**2.27**	910	54	1.33	0.77	2.32	0.631
Fried	1625	172	1.30	0.98	1.74	929	124	1.27	0.90	1.79	696	48	1.38	0.84	2.28	0.777
Stewed	2072	228	**1.71**	**1.19**	**2.47**	1143	167	**1.97**	**1.26**	**3.08**	929	61	1.26	0.68	2.33	0.253
Oven-Baked	1654	179	**1.62**	**1.20**	**2.20**	918	136	**2.03**	**1.41**	**2.93**	736	43	1.01	0.61	1.66	**0.022**
** Red Meat**	** *n* ** **= 2740**	** *n* ** **= 269**				** *n* ** **= 1546**	** *n* ** **= 195**				** *n* ** **= 1194**	** *n* ** **= 74**				
Griddle/BBQ	2395	237	**1.59**	**1.03**	**2.45**	1345	169	1.43	0.88	2.34	1050	68	2.15	0.90	5.17	0.409
Fried	1809	194	1.27	0.94	1.72	1039	141	1.23	0.86	1.77	770	53	1.36	0.79	2.34	0.761
Stewed	2392	247	**1.62**	**1.01**	**2.60**	1353	180	1.68	0.95	3.00	1039	67	1.49	0.66	3.35	0.805
Oven-Baked	1418	136	0.95	0.72	1.25	824	103	0.99	0.72	1.38	594	33	0.86	0.52	1.40	0.616

^a^ OR: odds ratio; LL: lower limit of the 95% confidence interval; UL: upper limit of the 95% confidence interval, *p*-Int: *p*-value for the interaction by sex. ^b^ Binary logistic regression adjusted by sex, age, education, family history of stomach cancer, physical activity (METs), smoking, BMI and energy, alcohol, fruits, salty fish, olives, and type-specific meat intake as fixed effects terms and province of residence as a random effect. ^c^ Binary logistic regression adjusted by age, education, family history of stomach cancer, physical activity (METs), smoking, BMI and energy, alcohol, fruits, salty fish, olives, and type-specific meat intake as fixed effects terms, province of residence as a random effect and including an interaction with sex. ^d^ Binary logistic regression adjusted by sex, age, education, family history of stomach cancer, physical activity (METs), smoking, BMI and energy, alcohol, fruits, salty fish, olives and other type-specific meat cooking methods as fixed effects terms and province of residence as a random effect. Non-consumers of the corresponding meat type were excluded. Risk for consumers of type-specific meat and cooking method vs. non consumers. ^e^ Binary logistic regression adjusted by age, education, family history of stomach cancer, physical activity (METs), smoking, BMI and energy, alcohol, fruits, salty fish, olives, and other type-specific meat cooking methods as fixed effects terms, province of residence as a random effect and including an interaction with sex. Non-consumers of the corresponding meat type were excluded. Risk for consumers of type-specific meat and cooking method vs. non consumers.

**Table 6 nutrients-14-04852-t006:** Association between gastric adenocarcinoma incidence and meat-type specific doneness preference and cooking methods by tumor subsite (cardia, non-cardia) and morphology (intestinal, diffuse) (restricted to consumers of each type of meat).

		Tumor Subsite									Tumor Morphology								
	**Controls**	**Cardia**				**Non-Cardia**					**Intestinal**				**Diffuse**				
	** *n* **	** *n* **	**RRR ^a,b^**	**LL ^a,b^**	**UL ^a,b^**	** *n* **	**RRR ^a,b^**	**LL ^a,b^**	**UL ^a,b^**	***p*-Het ^a^**	** *n* **	**RRR ^a,b^**	**LL ^a,b^**	**UL ^a,b^**	** *n* **	**RRR ^a,b^**	**LL ^a,b^**	**UL ^a,b^**	***p*-Het ^a^**
**Doneness Preference**																			
** White Meat**	**2520**	**62**				**175**					**94**				**55**				
Rare/Medium	1740	41	1			99	1			0.233	51	1			37	1			0.185
Well-done	780	21	1.08	0.62	1.86	76	**1.57**	**1.14**	**2.16**		43	**1.69**	**1.10**	**2.59**	18	1.05	0.59	1.87	
** Red Meat**	**2582**	**63**				**183**					**101**				**57**				
Rare/Medium	2067	48	1			132	1			0.842	70	1			48	1			**0.051**
Well-done	515	15	1.33	0.72	2.44	51	**1.42**	**1.00**	**2.02**		31	**1.61**	**1.02**	**2.53**	9	0.70	0.34	1.44	
	**Controls**	**Cardia**				**Non-Cardia**					**Intestinal**				**Diffuse**				
	**n**	**n**	**RRR ^a,c^**	**LL ^a,c^**	**UL ^a,c^**	**n**	**RRR ^a,c^**	**LL ^a,c^**	**UL ^a,c^**	***p*-Het ^a^**	**n**	**RRR ^a,c^**	**LL ^a,c^**	**UL ^a,c^**	**n**	**RRR ^a,c^**	**LL ^a,c^**	**UL ^a,c^**	***p*-Het ^a^**
**Cooking Methods**																			
** White Meat**	**2711**	**65**				**197**					**104**				**66**				
Griddle-grilled/BBQ	1961	46	1.00	0.56	1.79	136	1.12	0.80	1.58	0.729	69	1.15	0.73	1.81	42	0.69	0.41	1.19	0.151
Fried	1625	46	1.45	0.81	2.59	121	1.07	0.78	1.48	0.357	69	1.44	0.92	2.26	41	0.97	0.57	1.65	0.251
Stewed	2072	53	1.20	0.62	2.30	168	**1.73**	**1.14**	**2.63**	0.340	92	**2.40**	**1.28**	**4.49**	52	1.02	0.55	1.88	**0.051**
Oven-Baked	1654	50	**2.14**	**1.15**	**3.96**	127	1.36	0.98	1.88	0.192	67	**1.57**	**1.01**	**2.44**	47	1.55	0.88	2.72	0.972
** Red Meat**	**2740**	**65**				**197**					**106**				**64**				
Griddle-grilled /BBQ	2395	52	0.53	0.27	1.05	178	**1.71**	**1.02**	**2.85**	**0.005**	95	**2.16**	**1.09**	**4.29**	55	0.86	0.41	1.82	0.069
Fried	1809	51	1.57	0.84	2.93	138	1.09	0.79	1.52	0.305	76	1.25	0.79	1.97	45	0.98	0.56	1.73	0.511
Stewed	2392	61	1.63	0.58	4.61	179	1.32	0.79	2.21	0.721	98	1.82	0.86	3.86	59	1.39	0.54	3.57	0.655
Oven-Baked	1418	31	0.70	0.41	1.17	102	1.00	0.74	1.35	0.223	57	1.15	0.76	1.73	37	1.23	0.73	2.07	0.833

^a^ RRR: relative risk ratio; LL: lower limit of the 95% confidence interval; UL: upper limit of the 95% confidence interval, *p*-Het: *p*-value for the heterogeneity of effects; BBQ: barbequed. ^b^ Multinomial logistic regression adjusted by sex, age, education, family history of stomach cancer, physical activity (METs), smoking, BMI, province or residence and energy, alcohol, fruits, salty fish, olives, and type-specific meat intake. Non-consumers of the corresponding meat type were excluded. ^c^ Multinomial logistic regression adjusted by sex, age, education, family history of stomach cancer, physical activity (METs), smoking, BMI, province of residence and energy, alcohol, fruits, salty fish, olives, and other type-specific meat cooking methods intake. Non-consumers of the corresponding meat type were excluded. Risk for consumers of type-specific meat and cooking method vs. non consumers.

## Data Availability

The datasets generated for this paper are not publicly available due to restrictions imposed by the Carlos III Committee for Ethical Research, but are available from the principal investigator on reasonable request.

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
