# Peer review of "Meat Intake, Cooking Methods, Doneness Preferences and Risk of Gastric Adenocarcinoma in the MCC-Spain Study"

_nutrients, 2022, doi:10.3390/nu14224852_

Round 1

Reviewer 1 Report

The aim of this study is to evaluate in the MCC-Spain multicontrol study the role of meat intake in the incidence of adenocarcinoma of the stomach.

The authors also assessed the cancer risk correlated with meat type, cooking methods and cooking preferences, overall and by histological subtypes and anatomical sub-sites.

The data are interesting and well presented. The study has good potential but there are shortcomings that prevent its publication in this form. 

1. The data is very old, dating from 2008 - 2023. The authors have already published the data in another paper:  https://link.springer.com/article/10.1007/s10120-017-0774-x

2. There are many plagiarised parts that should be corrected - - see attached file.

3. What biochemical aspects are involved in the tumour pathogenesis of red meat consumption and stomach cancer?

4. What is the role of fibre in the prevention of this type of cancer?

5. Preventive effect of the Mediterranean diet and negative effect of diets based on meat consumption (e.g. ketogenic). 

Author Response

Reviewer 1. Comments and Suggestions for Authors

The aim of this study is to evaluate in the MCC-Spain multicontrol study the role of meat intake in the incidence of adenocarcinoma of the stomach.

The authors also assessed the cancer risk correlated with meat type, cooking methods and cooking preferences, overall and by histological subtypes and anatomical sub-sites.

The data are interesting and well presented. The study has good potential but there are shortcomings that prevent its publication in this form. 

  1. The data is very old, dating from 2008 - 2023. The authors have already published the data in another paper:  https://link.springer.com/article/10.1007/s10120-017-0774-x

As the reviewer affirm, the recruitment of the MCC-Spain project was carried out between September 2008 and December 2013. The protocol of the project was followed in 23 hospitals from 12 Spanish provinces, constituting the largest population-based multicase-control study developed in Spain until that moment. This large study was intended to evaluate the influence of environmental exposures and their interaction with genetic factors in five of the most common tumours in Spain (prostate, female breast, colorectal, gastric cancer and chronic lymphocytic leukaemia).

Due to the comprehensive information available on cases and controls, there are many sub-studies being performed at present.  We think that, when the data are good enough, it is positive we continue analyzing the information available. Even though dietary habits may have to some extent changed in the last 10 years, we consider that the estimates of the association between the exposure, i.e. meat intake, and gastric adenocarcinoma are still valid and applicable to the current situation.

The objective of the paper mentioned by the reviewer (https://link.springer.com/article/10.1007/s10120-017-0774-x) was to explore the association of three previously identified dietary patterns with gastric adenocarcinoma by sex, age, cancer site, and morphology, but no data related specifically to meat intake were analized nor showed in it. Although both papers evaluate the role of diet in gastric adenocarcinoma etiology, there are serious differences we want to highlight. 

Meat cooking practices and doneness preferences were not addressed in the previously mentioned paper we published in 2018. Besides, meat cooking practices and doneness preferences have been little studied, although these practices could independently increase the risk of GAC. We hypothesized that by studying the role of cooking practices and doneness preferences we might help explaining the currently observed heterogeneity of results among epidemiological studies.

In this manuscript, we aim to investigate the role of meat intake in the incidence of gastric adenocarcinoma, including type of meat, meat cooking methods and doneness preferences, overall and by histological subtypes and anatomical subsites, within the MCC-Spain Multicase-control study. These results have not been published in the previous manuscript.

  1. There are many plagiarised parts that should be corrected - - see attached file.

Thank you for this comment. As indicated, we have thoroughly reviewed the manuscript to correct this point.

  1. What biochemical aspects are involved in the tumour pathogenesis of red meat consumption and stomach cancer?

Several biochemical mechanisms have been proposed as potentially involved in the carcinogenic effect of red meat. Meat components highlighted as potentially related to tumour pathogenesis in the monograph about red and processed meat published by the International Agency for Research on Cancer (IARC) in 2018 are: haem iron, lipid oxidation products, heterocyclic aromatic amines, polycyclic aromatic hydrocarbons, and N-Nitroso compounds.

In the fourth paragraph of the Discussion section, we mentioned some of these mechanisms, such as the potential of iron contained in red and processed meat to damage DNA through the production of reactive oxygen species and the formation of N-Nitroso compounds, as well as to favour H. pylori survival and growth. In addition, saturated fats, cholesterol and polycyclic aromatic hydrocarbons present or generated when cooking red meat may contribute to the carcinogenic effect. In the Discussion section, we have added more detail addressing the mechanisms that may be involved in the role of meat consumption in the etiopathogenesis of gastric cancer.

  1. What is the role of fibre in the prevention of this type of cancer?

Thank you for your question. Recently, as regards to fibre, the last WCRF/AICR report on Diet, nutrition, physical activity and stomach cancer stated that the role of fibre in the prevention of gastric cancer is limited. Specifically, the report affirms: “The following exposures, which were also previously too limited to draw conclusions in the Second Expert Report and not updated as part of the CUP Stomach SLR 2015 due to a lack of new evidence, remained ‘limited – no conclusion’: cereals (grains) and their products, dietary fibre, potatoes, starchy roots, tubers and plantains, nuts and seeds, herbs, spices and condiments, milk and dairy products, fats and oils, total fat, fatty acid composition, cholesterol, sugars, sugar (sucrose), fruit juices, dietary nitrate and nitrite, N-nitrosodimethylamine, drying or dried food, protein, thiamin, riboflavin, vitaminC, vitamin D, multivitamin/mineral supplements, calcium, iron, selenium supplements, carotenoids, culturally defined diets, meal frequency, eating speed and energy intake.” Then, as far as we know, evidence about the association of fibre intake with Gastric cancer is too limited to draw firm conclusions at present.

  1. Preventive effect of the Mediterranean diet and negative effect of diets based on meat consumption (e.g. ketogenic). 

We also thank the reviewer to mention this topic, since despite the numerous studies exploring the effect of the Mediterranean diet in GC risk, the  WCRF/AICR still considers the evidence insufficient to draw firm conclusions (https://www.wcrf.org/wp-content/uploads/2021/02/Summary-of-Third-Expert-Report-2018.pdf). Although some studies in animals suggest that ketogenic diet might inhibit tumor growth (Khodadadi  2017[1]), its effect in the prevention of cancer, in general and specifically in gastric cancer, is not clear (Shah 2022[2], Lane 2021[3] ). Nonetheless, we expect this type of diet to be very infrequent in the MCC-Spain participants, limiting the capacity to analyze its potential effect.

[1] Khodadadi S, Sobhani N, Mirshekar S, Ghiasvand R, Pourmasoumi M, Miraghajani M, Dehsoukhteh SS. Tumor Cells Growth and Survival Time with the Ketogenic Diet in Animal Models: A Systematic Review. Int J Prev Med. 2017 May 25;8:35.

[2] Shah UA, Iyengar NM. Plant-Based and Ketogenic Diets As Diverging Paths to Address Cancer: A Review. JAMA Oncol. 2022 Aug 1;8(8):1201-1208. 

[3] Lane J, Brown NI, Williams S, Plaisance EP, Fontaine KR. Ketogenic Diet for Cancer: Critical Assessment and Research Recommendations. Nutrients. 2021 Oct 12;13(10):3562.

Reviewer 2 Report

I congratulate the Authors performing an interesting and sound study

Author Response

Reviewer 2. Comments and Suggestions for Authors

I congratulate the Authors performing an interesting and sound study

We thank the reviewer for the evaluation of the manuscript and their comments.

Reviewer 3 Report

The study was well designed and conducted. Findings are clearly presented in the tables and discussed in the text. The manuscript needs to be checked for a few typos. Overall this is a well put together manuscript.

Specific Comments:

1. What is the rationale for including olives in the baseline characteristics? Extra virgin olive oil consumption might be a better assessment. It would be helpful to include a healthy eating index in the baseline characteristics, such as a Mediterranean diet score or the alternative healthy eating index.

2. In the limitations it should more clearly stated that the significant differences (e.g. age, sex, and physical activity) in the control vs. cases populations may have confounded the findings.  

3. The Discussion could use more discussion of the contrast in findings between males and females.

Author Response

Reviewer 3. Comments and Suggestions for Authors

The study was well designed and conducted. Findings are clearly presented in the tables and discussed in the text. The manuscript needs to be checked for a few typos. Overall this is a well put together manuscript.

We thank the reviewer for the examination of the manuscript and their comments. We have reviewed the text to address the specific comments.

Specific Comments:

  1. What is the rationale for including olives in the baseline characteristics? Extra virgin olive oil consumption might be a better assessment. It would be helpful to include a healthy eating index in the baseline characteristics, such as a Mediterranean diet score or the alternative healthy eating index.

We understand the reviewer confusion about the inclusion of olives as a possible confounder. As the reviewer might know, there is firm evidence that links the intake of foods preserved in salt, including pickled vegetables, with gastric cancer. Olives are highly consumed in Spain as an appetizer and the olives are cured in salt. So we included them as a confounder to adjust for the possible effect of salt consumption taking into account that in many occasions in which meat is consumed (barbeques or diners with friends, meals in restaurants, etc.) olives consumption is also present.

  1. In the limitations it should more clearly stated that the significant differences (e.g. age, sex, and physical activity) in the control vs. cases populations may have confounded the findings.  

We thank the reviewer for addressing this point, since it was also a principal concern for us. Although controls were frequency-matched to cases, by age, sex and region, our multi-case control design hampered a perfect balance in age, sex or region among a specific tumour cases and controls. To reduce the possibility of confounding derived from the imbalance between cases and controls (in age, sex and other gastric cancer risk factors, including physical activity), we used multivariable analysis adjusted by age and sex, and including region as a random effect term. A sentence about this point has been added in the limitations subsection of the discussion section.

  1. The Discussion could use more discussion of the contrast in findings between males and females.

We thank the reviewer for this last comment. Although the number of GAC cases among women was small, resulting in lower statistical power to detect possible associations, we also think that different findings between males and females need to be emphasized. Accordingly, we have modified the discussion section to highlight this point.

Round 2

Reviewer 1 Report

The authors addressed all my concerns